# A Conceptual Model Depicting How Children Are Affected by Parental Cancer: A Constructivist Grounded Theory Approach

**DOI:** 10.3390/children10091507

**Published:** 2023-09-04

**Authors:** Elise S. Alexander, Georgia K. B. Halkett, Blake J. Lawrence, Moira O’Connor

**Affiliations:** 1Discipline of Psychology, School of Population Health, Faculty of Health Sciences, Curtin University, Bentley, WA 6102, Australia; elise.alexander@uwa.edu.au (E.S.A.); blake.lawrence@curtin.edu.au (B.J.L.); 2Curtin School of Nursing/Curtin Health Innovation Research Institute (CHIRI), Faculty of Health Sciences, Curtin University, Bentley, WA 6102, Australia; 3Discipline of Psychology, School of Population Health/Curtin Health Innovation Research Institute (CHIRI), Faculty of Health Sciences, Curtin University, Bentley, WA 6102, Australia; m.oconnor@curtin.edu.au

**Keywords:** parental cancer, psychosocial, qualitative, children, family

## Abstract

Cancer patients’ children are vulnerable to psychosocial and behavioural issues. The mechanisms underlying how children are affected by their parent’s diagnosis are unknown, warranting further research. This study investigated how children are affected by their parent’s cancer diagnosis and provides a theoretical model conceptualising this experience. Informed by methods of grounded theory, embedded within a social constructivist framework, 38 informants (15 health professionals (HPs); 11 parents; 12 children (5 to 17 years)) were interviewed using a semi-structured format. Three themes were identified: (i) children were worried and distressed because they felt alone, (ii) parents were unable to tend to children’s needs because they were overwhelmed by practical factors, and (iii) HPs were not detecting children due to barriers that affected their visibility in clinical settings. The proposed Alexander’s Children’s Cancer Communication (ACCC) Model and clinical recommendations made can be used to guide the clinical practice and development of future intervention research.

## 1. Introduction

Globally, the number of dependent children living with a parent with a cancer diagnosis remains unreported. However, estimates in the United States indicate that in 2010 there were approximately 2.85 million children aged 18 years and under whose parent had been diagnosed with cancer [1]. In Australia, there are currently no population data regarding the prevalence and characteristics of children living with parental cancer, though a recent longitudinal study conducted in Western Australia (WA) reported that between 1982 and 2015, 25,901 children (approximately 24% of those between 0–11 years) experienced a parent’s diagnosis of cancer [2]. The five-year survival rates among Australian patients with cancer aged 25 to 49 years are rising [3], meaning that patients and their families, including patients’ children, are living longer with the impact of a cancer diagnosis [4]. Consequently, many of these patients will be supporting dependent children while also coping with their diagnosis [5,6,7], presenting a major challenge for this cohort.

Research indicates the overall adjustment and emotional wellbeing of patients’ children is negatively affected, rendering children at risk of developing various maladaptive psychosocial, emotional, and behavioural stress responses [8,9]. These include somatic complaints [10], separation anxiety, high levels of distress, confusion, rumination, worry and intrusive thoughts [11,12,13]. While these negative symptoms may dissipate over time for some children, many children are likely to remain vulnerable to long-term problems, including self-injury and post-traumatic stress symptoms [14,15,16]. Despite these negative effects, there are also reports of resilience building and the potential for post-traumatic growth among cancer patients’ children [9,13,17]. Age, gender, cancer stage, pre-existing comorbidities, parent’s marital status and psychological health are factors which appear to influence how children are affected [8,9,18,19,20]. However, it remains uncertain what underlying mechanisms may affect children’s capacity to cope and adjust to their parent’s cancer diagnosis.

Parents have indicated they are uncertain how to support children and seek clinical or therapeutic help [21], and it has been identified that children are undetected in clinical and health systems and typically not on the radar of the patient’s oncological treating team [22,23]. Compounding these issues, parents are unlikely to address concerns about their children with health professionals (HPs) and HPs often do not discuss the subject of children with patients [21]. Yet, children prefer to be supported by parents, and HPs are well-placed for detecting distress and referring children to appropriate services [21]. Consequently, children do not appear to be receiving the help they need and are often left to support themselves [24,25]. Much of the literature focuses on supporting patients and their spouses, with little consideration of the needs of patients’ children [26,27].

In our systematic review of the literature [9], which preceded and informed the current study, we found there to be limited reporting of psychosocial interventions for supporting patients’ children’s coping and wellbeing, in the empirical literature. Our findings in this review are supported by other similar reviews; see reviews by the authors of [11,28]. Participants’ qualitative feedback reported in intervention studies supports their feasibility and acceptability, indicating there is a need for interventions among parents and children [28]. However, recent systematic reviews indicate interventions are not effective in mitigating children’s various psychosocial and behavioural outcomes, including depression and anxiety [28], and there is demand for more empirically developed and rigorously evaluated interventions [9,11,29]. Interventions also lack a theoretical framework or model that conceptualises the mechanisms which may explain how children are affected by their parent’s diagnosis [28,29]. A new robust theoretical model should therefore be developed, and subsequently used to inform any future intervention studies designed to improve psychosocial and behavioural outcomes among children experiencing their parent’s cancer diagnosis. The present study aimed to explore how children are affected by their parents’ cancer diagnosis, from the perspectives of children, parents, and health professionals, to inform the development of a new theoretical model.

## 2. Materials and Methods

An integrated theoretical approach combining symbolic interactionism [30] and Ecological Systems Theory (see Figure 1) [31] was used to provide the theoretical lens for developing an explanatory model conceptualising how children are affected by their parent’s cancer diagnosis. This approach has been used in other studies investigating children’s social experiences (e.g., [32,33]) and was deemed appropriate for the purposes of this study as it accounts for the high level of influence children’s social interactions and the environment has on their developmental outcomes when a parent is diagnosed with cancer.

Constructivist grounded theory informed the methodology and methods used in this study [35,36]. This methodology aims to develop a detailed understanding of a person’s unique social experiences [35,36]. It adopts flexible research methods and recognises the person’s construction of reality is a co-constructive process that occurs between the participant and the researcher. Such adaptive methods are important among diverse populations, particularly children [37,38]. Additionally, rather than a simple description of events, this methodological approach enabled us to produce a theoretical conceptualisation of how children are affected by their parent’s cancer diagnosis [35,36].

### 2.1. Ethics Approval

Sir Charles Gairdner and Osborne Park Health Care Group (SCGOPHCG) Human Research Ethics Committee (HREC) (approval number: 12102016; date of approval: 12 October 2016) ethics approval and Curtin University Human Research Ethics Committee (HREC) reciprocal ethics approval were both received for this study.

### 2.2. Participant Recruitment

Three key informant groups were recruited for this study: HPs, parents, and children. Health professionals were initially recruited through a Comprehensive Cancer Centre at a metropolitan tertiary teaching hospital in Perth, Western Australia. HPs within the centre were approached via one of the researchers (MOC) contacts and asked to advise potential participants of the study. Fliers were also posted on bulletin boards within the hospital and centre. Please see Table 1 for participant inclusion criteria. Potential participants either contacted the primary researcher directly or were contacted by the researcher via email. A mutually convenient time and location was then arranged. Participation was voluntary. Participants were provided with an information sheet and signed a consent form prior to the interview. Children of recruited parents were also given the option to be interviewed and their interviews were conducted independently of their parents. Data analysis and recruitment occurred concurrently. As new themes emerged, theoretical sampling was used to explore these further by approaching participants considered to have insight regarding novel themes [35,36,39]. Recruitment, interviews, and analysis continued until the data were considered rich and detailed, and no new categories were emerging, therefore indicating data saturation [35,40].

### 2.3. Participants

A total of 48 participants were interviewed between April 2017 to June 2018. This included 15 HPs, 11 patients and spouses of patients, and 12 children. HPs were predominantly female (80%) with a mean age of 51.46 (±10.5) years. Parents were also mostly female (90%) with a mean age was 39.7 (±7.44) years. Forty-five percent of parents were spouses of a parent with a cancer diagnosis. Just over half of children participants were female (58%) with a mean age of 9.2 (±3.5) years. See Table 2 for a summary of participants’ demographics.

### 2.4. Interviews

All interviews were conducted by the primary researcher (EA, female, Ph.D. candidate and experienced interviewer). Interviews were semi-structured, which enabled topic consistency while also promoting alternative lines of enquiry, further explanation, and examples of topics where relevant. This also encouraged participants to provide their own unrestricted perspectives. The focus of interview questions was to explore how participants perceived patients’ children were affected by their parent’s cancer diagnosis. The interview schedules were guided by the research question, a general literature review, and findings from a systematic review [28]. Interview schedules are documented in the relevant publications (for HPs, see [41]; for parents, see [42]; and for children, see [43]) or see Appendix A. Interview durations were as follows: HPs: 43 min (M = 42.69 (±22) range: 14.32–82.52); parents: 46 min (M = 45.88 (±10), range: 27.36–73.14); and children: 38 min (M = 37.13 (±22), range: 17.57–71.44).

### 2.5. Children’s Activity

A novel approach was developed and used during the children’s interviews to facilitate their capacity to articulate their responses to interview questions. This was influenced by drawing- and arts-based approaches or techniques [44,45] which are used in participatory research within vulnerable child populations such as those affected by sexual abuse [46,47] and war [48,49], often due to their ability to promote recall [47]. For more details regarding this approach, please refer to [43].

### 2.6. Data Collection

Individual HP interviews took place in person in the participant’s workplace (*n* = 8) or via a scheduled telephone call (*n* = 7). Individual parent interviews were conducted in the participants’ homes (*n* = 4), at Curtin University (*n* = 3), temporary accommodation (*n* = 2), the place of work (*n* = 1), or the tertiary hospital (*n* = 1). Children’s interviews were conducted in the same location as their parents were; however, parents were not in the room. Informed consent was received from all participants. Children were provided developmentally appropriate information sheets and verbal assent was attained per the National Statement on Ethical Conduct in Human Research chapter 4.2 (National Health Medical Research Council [NHMRC], [50]). Observational notes and journalling were used to record notable details regarding context and behaviours. Pseudonyms were used for all children’s names to ensure confidentiality and anonymity. For more details regarding the data collection processes, please refer to the relevant publications (for HPs, see [41]; for parents, see [42]; and for children, see [43].

### 2.7. Data Analysis

Interviews were digitally recorded and transcribed verbatim. Using methods of constructivist grounded theory, transcribed data were analysed using initial line-by-line coding of the first five transcripts focusing on gerunds (actions and processes) to identify codes. Data and codes were then transferred to Microsoft Excel to index the data into manageable chunks and develop preliminary themes. Transcripts were iteratively reviewed by members of the research team, and themes were discussed and refined. Data and themes were then transferred to a qualitative data analysis computer software package, NVivo12 (QSR International https://support.qsrinternational.com/s/, accessed on 29 June 2023), where the remaining transcripts were coded, while the researcher remained open to identifying further themes. Methods of constant comparison were used to elevate themes and develop higher-order categories [35,36]. Memoing techniques were also used to support themes to be moved from the descriptive to the analytical level [35,39]. For more details regarding the data analysis process. please refer to the relevant publications (for HPs, see [41]; for parents, see [42]; and for children, see [43]. Reference to guidelines and criteria outlined by Mays and Pope [51] and Braun and Clarke [52] promoted the rigour of analysis. The consolidated criteria for reporting qualitative research (COREQ) guidelines [53] were also used to promote study quality and reporting rigour.

## 3. Findings

In this study, the perspectives of children, parents, and HPs were explored to understand how children are affected by a parent’s cancer diagnosis. Three major themes were identified: (i) children were worried and distressed because they felt alone, (ii) parents could not tend to children’s needs because they were overwhelmed by the practical factors of a cancer diagnosis, and (iii) HPs were not detecting children’s distress because there were barriers which affected children’s visibility. Central to these three themes was the overarching concept that when a parent was diagnosed with cancer, their children were not seen and not heard. These themes are elaborated on and details of a proposed model that explicates these findings are presented below [35].

### 3.1. Children

#### Worried, Distressed, and Alone

Following a parent’s cancer diagnosis, children experienced heightened levels of ongoing worry and distress. While this worry and distress varied for children based on factors such as their age and personality traits, for all children this was greatly influenced by circumstances such as the patient’s (parent’s) health status, hospitalisation, and changes to treatment plans.


*Interviewer: “You feel worried and frustrated a lot of time?”*



*Child: “Yes”.*



*Interviewer: “When do you feel happy?”*


*Child: “When Mummy’s okay and she’s doing stuff”* (Batari; female: 8.5 years).

This worry and distress was very intrusive for children, and consequently children were often hypervigilant about their parent’s health and any changes; *“I just want to go home every day... Because I want to stay with my Mummy to make her feel better”* (Arianna; female: 6.5 years). Children whose parent was an outpatient appeared more vulnerable as they were constantly exposed to, and unable to escape from, the disease and treatment side-effects.

*“Her [patient] feet look bad. When she walks, her feet hurt. That’s why she always has her slippers on. But then they make a weird noise at the night. I can’t sleep and [sister] can’t sleep”* (Arianna; female: 6.5 years). Even after the parent was in remission, their children continued to experience ongoing levels of heightened worry and distress; *“I don’t think it (worry) had much of an effect on me then. I feel like it has more of an effect on me now because I know that my Mum survived and what could have happened and what she did to keep it all [life] going”* (Lucas; male: 12 years). This persistent worry and distress was similarly observed in children who were bereaved; *“I worry about the dogs dying. I worry about Mum dying. I worry about all of my family, really”* (Kayla; female: 10.5 years; her parent had died). This participant also evidenced these worries and concerns as their most prevalent worries in their drawing (Figure 2).

However, children did not appear to have much support to help them cope with this worry and distress. Most preferred to be supported by parents, however felt a loss of quality time with parents.


*Child: “I spent more time with her before she got sick. She’s having another operation to take the bag away and then we’re going to have more time to be with her again”.*



*Interviewer: “Are you looking forward to that?”*


*Child: “I’ve been waiting for it for 1000 years”* (Arianna, female: 6.5 years).

They were also conscious of not burdening their parents who they recognised were already struggling with the emotional, physical, and practical aspects of the cancer diagnosis; *“When she got cancer, she couldn’t do it [cooking] so she just gives us like baked beans or spaghetti”* (Darius, male: 13 years). They also felt alone and disconnected from other supports they would normally have, such as their extended family, friends, and school and sports communities; *“I miss my friends because most of them are not at home and we’re not going to school”* (Arianna, female: 6.5 years). Most children had questions about their parent’s cancer diagnosis; however, while children preferred to speak to their parents about these, *“I would ask my Mum, or I could ask my Dad. I would ask either one of them or maybe someone who was in the house”* (Sarah; female: 11 years). Their limited understanding and lack of sufficient knowledge about their parent’s cancer diagnosis was evident, which suggested this might not be occurring.


*Interviewer: “Can you tell me what you know about mum’s cancer?”*



*Child: “Brain cancer, kills people” (notably, the parent did not have a brain cancer diagnosis).*



*Interviewer: “Did someone you know have brain cancer?”*


*Child: “It’s Granddad. He died”* (Arianna; female: 6.5 years).

Many children felt they needed to talk to someone, however, considered their friends were unlikely to understand and they found it difficult to talk to adults, including their parents. Most children found it challenging to comprehend their complex thoughts and emotions and articulate these to adults, indicating their need for help with this.


*Interviewer: “Do you know what you would ask?”*



*Child: “I’m generally unsure of it, I just feel the need to know something”.*



*Interviewer: “Is it something you can ask Mum about?”*


*Child: “It might be, but I’m unsure of how to do this”* (Lucas, male: 12 years).

One child indicated they needed help to express their thoughts and feelings and that this was a great concern for them (Figure 3).

Another child who was reluctant to speak to the interviewer and whose parent assumed they had little awareness of their diagnosis was encouraged to demonstrate their knowledge of their parent’s diagnosis through drawing, whereby they correctly indicated the primary cancer site of the diagnosis (Figure 4).

Children were also unlikely to approach HPs for support, as they did not consider them in the first place and assumed HPs would not be able to help them, which was supported by the experiences of some children who found their engagement with HPs ineffective; *“The [child’s] therapist only goes for 10 min—she asks questions. I don’t really get a chance to ask*” (Batari; female: 8.5 years).

### 3.2. Parents

#### Children’s Needs Are Unattended

Parents were typically overwhelmed and burdened upon receiving the diagnosis which rendered them unable to tend to children’s needs. They were often dealing with their own shock and disbelief elicited by the cancer diagnosis, while endeavouring to comprehend vast amounts of complex technical information which they felt could be more effectively and supportively disseminated by HPs; *“The information comes so fast and quick; you’re making ‘informed decisions’… how can you make ‘informed decisions’ when information comes in that thick and fast and in things that we know nothing about”* (Parent 2). Consequently, while parents did their best to talk to children and answer any questions about the diagnosis, they were simultaneously trying to understand the information provided and work out what was happening *“We weren’t completely sure ourselves. So, we didn’t really want to tell the children [about the diagnosis]”* (Parent 4). As such, parents found it difficult to know how much to tell children; *“It’s probably more just trying to protect them… it’s hard to know how far you go and what you need to tell them”* (Parent 5).

Parents were also overwhelmed and burdened by the practical factors of a cancer diagnosis, including navigating complex healthcare systems,

*“If you have to phone to change an appointment, you don’t go to a receptionist. You go to some third party who may or may not be able to change an appointment or answer a question, so it’s absolutely hopeless whatever that system is, so we don’t even bother”* (Parent 1)

and implementing life changing decisions to accommodate diagnosis and treatment; *“We’re sitting there and we’re having to decide on the spot whether or not he’s going to have further treatment”* (Parent 7).

Often, the healthy parent was required to take on the ill parent’s role and responsibilities, *“She’s [wife] taken a lot of the load of stuff that I would have dealt with before. She deals with most of the financial stuff now; just trying to not get me stressed out so I can concentrate on getting better”* (Parent 4), and felt forced between being a parent to their children or being the patient’s advocate; *“To manage the illness—I felt I was put in the position of having to choose between parenting and supporting (patient) through to the end of his life”* (Parent 10). Alternatively, the ill parent felt divided between prioritising their health and the impact of the disease and the demands of treatment, with their responsibilities of being a good parent and partner; *“I still don’t think they [children] understand how tired and sick I am. Sometimes it gets overwhelming. They get quite demanding… so trying to explain to them but trying not to make them feel like I’m getting cross at them”* (Parent 1).

Parents found themselves challenged in maintaining a level of stability and normalcy for their children.


*“There’s always something that comes up. We’ve tried to plan for the best but expect the worst”.*


Often children’s activities, including sports, hobbies, social activities, and visiting friends and extended family were curtailed or stopped due to the demands and constraints of the diagnosis and treatment; *“We had them enrolled in gymnastics and basketball… everything was just a bit full on when I got sick… we decided to strip back all the things we were doing with the kids”* (Parent 3).

Significant upheaval was also unavoidable at times with some families needing to relocate to be closer to treatment options and change schools; *“I rang the Leukaemia Foundation; started packing the house. I changed the kids’ school; I got all that done within that week because the next week we had to be in Perth”* (Parent 5). This was particularly challenging for rural and regional families where relocation was not optional. Families where the patient was an out-patient faced unique challenges such as isolating and implementing extreme routines around hygiene and treatment particularly when the patient was neutropenic.

*“While he’s neutropenic you can eat an apple, but you’ve got to wash your hands; wash your face before you kiss or hug dad; you’ve just got to be really conscious. Lots of extra handwashing. We have good hygiene, but I’m just seeing bugs everywhere. It’s cleaning constantly”* (Parent 2).

Despite parents wanting to protect their children from being exposed to the physical, psychological, and mental detriments of cancer, this was often impossible, particularly when the patient was an out-patient,

*“The children pick up on my stress. They certainly pick up on his stress. My five-year-old was wetting the bed and the more [patient] got sleep deprived because he was getting up in the middle of the night changing sheets, the less tolerant of it he became and that becomes like a negative cycle”* (Parent 9),

the family had limited childcare support,

*“The first time I brought her to the hospital, she saw a lot of patients in very bad conditions”* (Parent 8),

or there was an emergency.

*“He [patient] got very sick and ended up in the high intensity unit because they had to call a code blue, which was probably a bit of a shock seeing Dad so sick in hospital, and for [child]—that was probably the hardest week for him. It was stressful for all of us; me trying to still work, going into hospital every night”* (Parent 4).

### 3.3. HPs

#### Children Are Invisible

There were many barriers that rendered children invisible among HPs which meant children were not being detected and referred on to the appropriate staff or services that could assist with this. For instance, there were no routines and standardised screening processes in place to systematically detect patients’ children and refer them on; therefore, HPs were relied on to remember and implement these processes themselves. However, this was not routinely occurring, and HPs typically relied on parent’s telling them they had children, rather than asking them about children.

*“That’s [screening] something we initiate ourselves. We normally try to get as much information about their social life and their family life as we can and then if they tell us they do have kids, then we explore”* (HP5).

Moreover, there was confusion around who was responsible for screening for patients’ children and subsequently children were typically overlooked as it was assumed someone else was doing this; *“If everyone is thinking the social worker will make a referral, it’s leaving it to the social worker, but they don’t automatically see every patient. So, there’s a lot that can be overlooked”* (HP11).

When considering approaching children, HPs were aware that conversations with children needed to be child-centric, that is, conversations needed to be timely and ongoing to accommodate children’s processing of information and developing cognitive sophistication.

*“Sometimes they [children] will process it through play therapy, but not be able to articulate it verbally how they’re feeling. Then, once the diagnosis has got to a safer distance, they might be able to engage in some verbal dialogue, or as they’re getting a little bit older, they might be in a position to articulate and want to revisit what’s happened”* (HP9).

However, most HPs felt they did not possess sufficient developmental knowledge or training to facilitate this level of communication with children. Consequently, HPs were reluctant to approach children as they felt they could potentially do more harm than good by opening up a conversation they were not equipped to have.

*“I’m not experienced in child psychology. I really am fearful that I would be doing an injustice opening up a conversation that I didn’t have the tools to complete”* (HP3).

Community services that were experienced with cancer patients’ children, and had resources and programs available to support them, were reliant on clinically based HPs to identify children, recall their services, and then refer or direct children to these services; *“We are relying on nurses and people in the healthcare profession to remember about [service]. It is not usually part of their standardised screening, but ultimately it would be good for it to be so that more referrals can come to [service]”* (HP11).

Yet, clinical HPs indicated their ongoing need for support with this to promote their recall and awareness of these services in the first instance, and to inform them of their current operational status, the services they provided, and their intake criteria. These HPs felt that it could be more detrimental to send children to a service that was no longer operational or not designed to support their needs; *“Probably there are a number of supports out there, but it’s having access to them and working out who can have access to those different things” … “More education and knowing the usual things of being able to signpost people to where help can be sought”* (HP7).

Exacerbating children’s lack of visibility among HPs were parents’ preferences to shield their children from hospital and clinical settings which limited HPs and children’s capacity to seek out one another.

*“I feel a lot of the time they shield the kids, so they don’t bring them in”* (HP4).

HPs also recognised that children were reluctant to talk to HPs, including those who were experienced with children and positioned to support them:

*“With the ones that are reluctant to talk, it is a lot more challenging, and I feel that even when I am trying to build more of a rapport with them and sneakily get some questions in here and there, it’s not always going to go well”* (HP12).

Furthermore, children typically wanted to protect their parents by concealing their thoughts and feelings from parents and HPs. Thus, this made it difficult to recognise when children were struggling; *“Sometimes there are ones that have never really spoken about it because they’re afraid of upsetting their parent or whatever the circumstance might be”* (HP12).

### 3.4. Explanatory Model: How Children Are Affected by Their Parent’s Cancer Diagnosis

Based on the perspectives of children, parents, and HPs, the overarching concept that cancer patients’ children were not seen and not heard emerged. When a parent was diagnosed with cancer, patients’ children were unlikely to receive the timely support needed to help them cope with their heightened levels of worry and distress, resultant from the diagnosis. This appeared to be due to the compromised nature of children’s interactions with parents and HPs. The mechanisms which underpinned the nature of these interactions were (i) a breakdown of communication among children, parents, and HPs, and (ii) barriers that prevented children, parents, and HPs from interacting with each other. These mechanisms are visually conceptualised in the proposed explanatory model titled Alexander’s Children’s Cancer Communication (ACCC) model (Figure 5).

#### Compromised Social Interactions and Communication Breakdowns

Children, parents, and HPs construct meaning through their social interactions with one another and their sharing of information [30]. However, our findings indicated fundamental social interactions were not occurring, particularly between children and parents, and children and HPs. When a parent was diagnosed with cancer, dramatic changes were observed in children’s micro-, meso, and exosystems which impacted the nature of these social interactions [31]. For instance, parents’ relationships with children were typically transformed and HPs were thrusted into children’s microsystems where their influence on children was directly felt. Additionally, children’s pre-existing supports such as extended family, friends, school, and sports communities became less proximal. These changes were often unsupported, and this presented barriers which challenged children’s meaningful interactions within these relationships.

## 4. Discussion

This study explored how children are affected by their parent’s cancer diagnosis from multiple perspectives. Patients’ children felt worried and distressed because they were alone in dealing with their complex thoughts, emotions, and experiences. Children’s needs were often unattended because parents were overwhelmed by the diagnosis and children’s visibility was reduced among HPs. These findings contribute to the growing body of parental cancer research by supporting current literature, indicating there is significant disruption experienced by children which impacts their psychosocial wellbeing [9,54]. Yet, parents’ parenting ability is often affected, and they want clinical help supporting children’s coping and developmental outcomes [21,55,56]. However, fears and barriers render HPs reluctant to broach the subject of patients’ children [21,55,57]. Our novel findings advance the literature by elucidating the underlying processes which contribute to these outcomes, and which have informed the development of our explanatory model.

Children felt confused about aspects of their parent’s diagnosis, which is commonly reported in the literature [9,58], suggesting their information and communication needs are not being met. Typically, when children are insufficiently informed, they attempt to make sense of the diagnosis on their own; however, this is often fraught with misconceptions and magical thinking which exacerbates fears and heightens feelings of worry and distress [55,56]. Children in this study found it difficult to comprehend and articulate their thoughts and feelings, which impacted their capacity to effectively communicate with adults, indicating they need help with this process. However, findings in the parental cancer literature suggest parents assume a higher communication quality than children do [59] and underestimate children’s information needs [60]. Furthermore, this reflects the developmental literature reporting that adults typically expect children to communicate at an adult level, and that the communication gap between adults and children needs to be more supported [61]. Open, timely, and age-appropriate communication is crucial to mitigating confusion and improving children’s coping and adjustment to the diagnosis [11,59,62].

Parents felt overwhelmed by and struggled to understand the vast amount of information received about the diagnosis, which impacted their capacity to communicate with children. These findings reflect the previous literature where parents commonly report their anxiety and uncertainty regarding how to provide ongoing and age-appropriate communication for children and are concerned for any harm they might cause [52,53,55]. Parents require clinician help with this process, yet this support is not routinely offered in healthcare [55,58]. HPs are reluctant to broach the subject of children with patients [21] and there are few empirically based resources [28,29]. Communication interventions are methodologically weak [28] and communication resources available through leading not-for-profit organisations are generally not empirically evaluated with sufficient rigour. Greater guidance and support are required to promote parent’s ability to discuss their cancer diagnosis with children to ensure children’s communication and information needs are met [9,53].

Parents were further challenged to tend to children’s information and communication needs due to being overwhelmed by the more practical and burdensome aspects of the cancer diagnosis. Parent–child communication is made increasingly difficult if parents are not mentally and physically capable. Similarly, to the current study, Sinclair and colleagues [56] found that the impact of diagnosis and treatment effected the feasibility and accessibility of communication resources among mothers with breast cancer, and therefore, warranted careful consideration for how communication information is disseminated to parents. Significant upheaval and disruption when a parent is diagnosed with cancer is documented in the literature [12,63]; however, our findings indicate that practical support issues among families remain unaddressed. Providing practical support around the challenges that come with a cancer diagnosis, such as the presence of a family support officer, would also enable parents to meet the communication and information needs of children.

Communication concerns with patients’ children were also evident among HPs who were reluctant to approach children. Aligned with the previous literature [21,55,57,64,65], HPs in this study felt they were not educated or experienced enough in talking with children, which impacted their confidence and willingness to seek out children. This concern is not unfounded, as a patient’s treating oncology team is primarily experienced in adult care and their interactions with children are minimal [21,55]. However, there is evidence indicating parent’s and children’s desire for HPs to be communicating medical and clinical based information with children [58,66,67]. There appears to be limited empirical resources available to assist HPs with communicating with children [55], except for (i) a newly proposed communication framework embedded within an e-learning intervention by Semple and McCaughan [68] designed to equip health professionals with supporting parents impacted by parental cancer and (ii) a communication tool developed and piloted by Hauken and Farbot [69] which aims to enhance communication with adolescents who are bereaved by or living with a parent with cancer. HPs are integral to supporting children’s communication and information needs; hence, there is need for further education, training, and access to resources for HPs to ensure this is routinely occurring.

Clinical barriers prevented HPs from routinely identifying patients’ children, yet HPs are well-positioned to be supporting children [57,58] and patients’ children will likely benefit from these discussions [11,67]. Consistent with our findings, barriers such as time [58,70,71], opportunity [55], and role [72] are commonly reported in the healthcare literature. In this study, children were also unlikely to approach HPs, consequently exacerbating the unlikeliness of HPs and children to interact and form meaningful relationships. Avoidance is common among children [73,74] which therefore places an emphasis on HPs to be actively seeking out and engaging with children [57]. Web-based interventions for healthcare support, information, and therapeutic treatment are proving increasingly useful among children in other areas such as mental health [75,76], and may address current problems experienced by HPs when supporting cancer patients’ children [57]. These interventions are generally accessible, tailorable, less taxing on resources (e.g., HPs time), and well-received by young people [76,77].

Breakdowns in communication and interaction are central components underlying how children are affected by parental cancer and therefore form the basis of our proposed model. The present study is the first to propose an empirically driven theoretical model that conceptualises how patients’ children are affected by their parent’s cancer diagnosis. One other study has proposed a causal model to potentially explain children’s adjustment to a parent’s cancer diagnosis [78]; however, this model was based on a review of the literature at that time. The model we propose, the ACCC model (Figure 5), provides a current explanation of how children are affected by their parents’ cancer diagnosis, by integrating the latest evidence within the literature and incorporating our research findings. To date, most interventions for cancer patients’ children have not produced clinically meaningful effects which may be attributed to the limited methodological rigour [29,79], including the absence of a theoretical foundation [28]. The ACCC model proposed in this study can be used to inform the development and implementation of future intervention strategies and therefore minimise the risk of Type III errors (i.e., rejection of the intervention when the intervention itself was inadequately designed and implemented ([80,81]). Furthermore, the ACCC model may be used to inform future clinical practice.

### 4.1. Clinical Recommendations

The clinical and practical implications and recommendations derived from this study, which promote the timely support of cancer patients’ children are as follows:Training and education aimed at developing HPs’ communication skills and developmental knowledge.The development of a communication tool to be used by HPs to effectively communicate with patients’ children. The new tool might also capture the benefits of technology and integrate the methods used in this study whereby children were encouraged to write and draw about their thoughts and feelings.The introduction of routine and standardised screening processes for HPs to detect patients’ children and efficiently refer them on to the appropriate supports and resources that are currently available. Oncology nurses may facilitate this approach while also being supported by the development of a new, or refinement of an existing, screening tool.The use of a multidimensional approach to support parents with the practical challenges of a cancer diagnosis. For example, a family support worker or social worker who can assist families from diagnosis onwards.

### 4.2. Limitations

There was a gender bias toward female HPs who participated in this study; however, this reflects the gendered nature of this industry (i.e., support professions) [1,56]. Male HPs may have had different perspectives; however, responses from participating male HPs were similar to those from females. HPs were recruited from one state (WA) within Australia. While findings are consistent with studies from other jurisdictions, it is acknowledged that generalisability to other healthcare systems may be limited. There was a gender bias toward mothers, which is common in the parental cancer literature [82]. Fathers may have presented different perspectives, and future studies need to explore their perspectives. Parents and children were recruited through opt in methods and predominantly through cancer support services and a Comprehensive Cancer Centre. Therefore, it is likely the participants in this sample had greater psychological awareness and health literacy. This is commonly reported in similar studies [83,84,85,86]; however, a sample from a more diverse background may be warranted. The age range of the children was broad; focusing on a narrower age range may have yielded different findings that reflect different experiences of children at different stages of development, and this could be considered in future studies.

## 5. Conclusions

Supporting cancer patients’ children is becoming an urgent area of concern in psycho-oncological research and care. Our findings indicate that when a parent is diagnosed with cancer, parents find themselves unavailable to tend to their children’s needs due to the demands associated with the cancer diagnosis, and HPs consider themselves challenged to provide the support that parents and children are asking for. Consequently, children feel they are not seen and not heard by parents and HPs, and experience ongoing levels of heightened worry and distress as they feel alone in dealing with the diagnosis. The underlying mechanisms contributing to these findings are breakdowns in communication among children, parents, and HPs, in addition to barriers that prevent them from interacting with each other. The findings from this study, including the proposed ACCC model which conceptualises how children are affected by their parent’s diagnosis, can be used to inform future research by providing a theoretical foundation to inform future intervention development and evaluation.

## Figures and Tables

**Figure 1 children-10-01507-f001:**
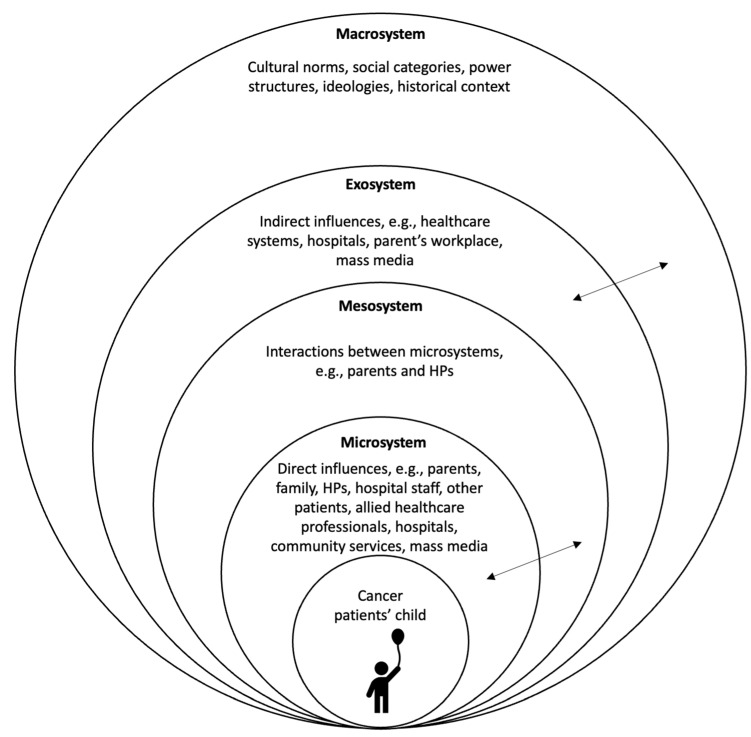
The ecology of children with a parent with cancer. Adaption of Bronfenbrenner’s (1979) Ecological Systems Theory model from “The Ecology of Cognitive Development” (pp. 3–44), by U Bronfenbrenner, in Development in Context: Acting and Thinking in Specific Environments, R. H. Wozniack & K. W. Fischer (Eds), 1993, Hillsdale, NJ: Erlbaum [34].

**Figure 2 children-10-01507-f002:**
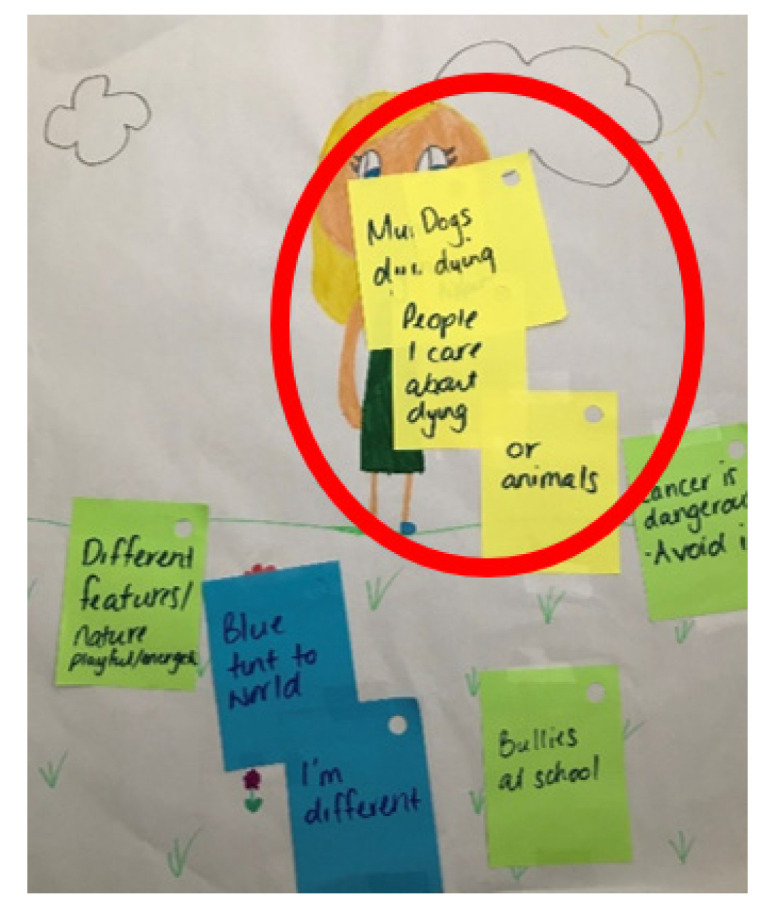
Child’s self-portrait and relative worries (Kayla; female: 10.5 years).

**Figure 3 children-10-01507-f003:**
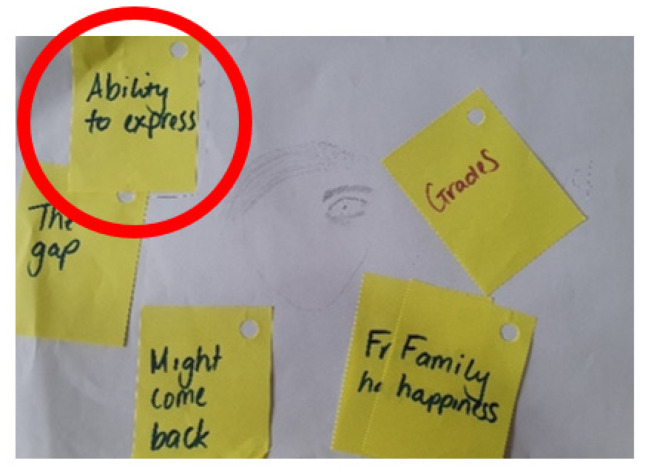
Child’s self-portrait and relative worries (Lucas; male: 12 years).

**Figure 4 children-10-01507-f004:**
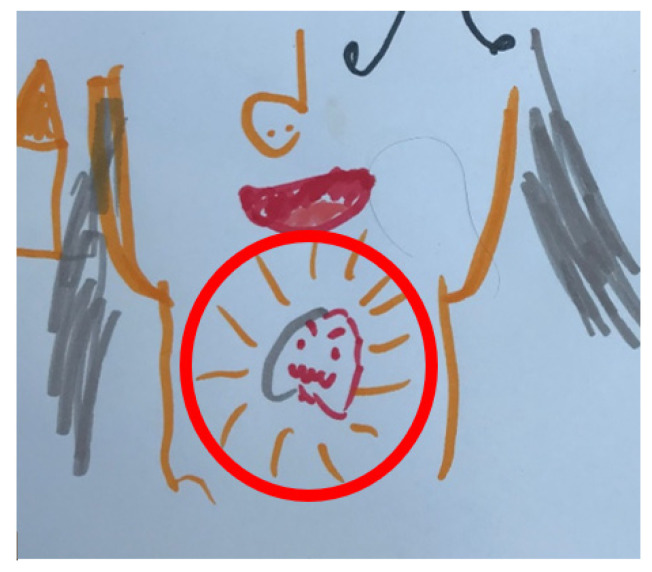
Child’s drawing of parent’s cancer site (James; male, 6 years old).

**Figure 5 children-10-01507-f005:**
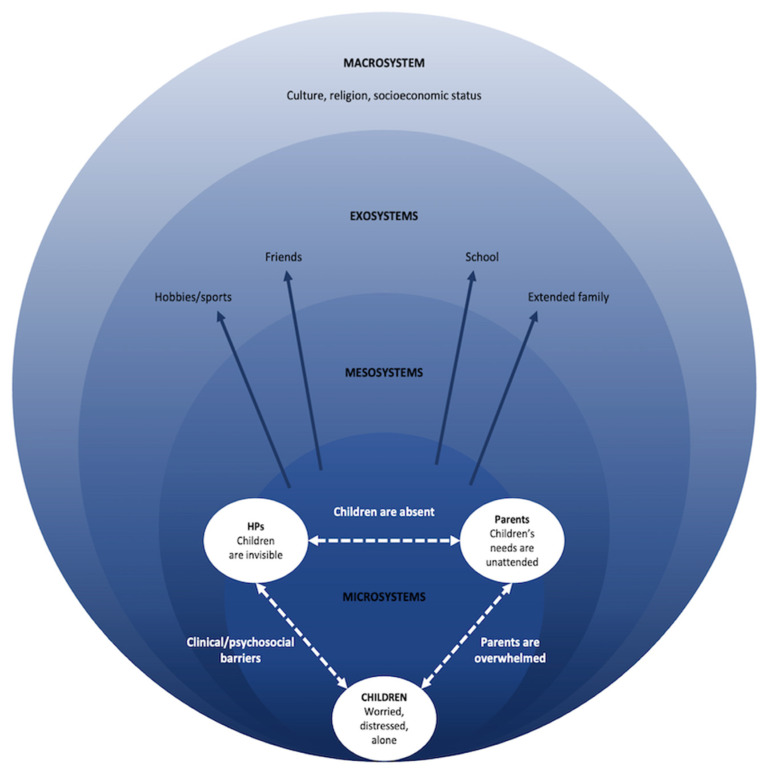
Explanatory model: Alexander’s Children’s Cancer Communication (ACCC) model.

**Table 1 children-10-01507-t001:** Participant Inclusion Criteria.

Population	Inclusion Criteria
HPs	Must be a health practitioner who is experienced in providing health care to patients with cancer who have a child or adolescent (up to the age of 18 years) living at home. For example, a Medical or Surgical Oncologist, Nurse Practitioner, Nurse Coordinator, Clinical Psychologist/Psychiatrist, Psychosocial support worker, or some other allied health specialist.
Parents	Must be a parent diagnosed with cancer or a parent whose partner has been diagnosed with cancer. Must have a child living with them aged 18 years or below.
Children	Must be living at home with a parent who has been diagnosed with cancer. Must be 18 years and under.

**Table 2 children-10-01507-t002:** Participant demographics.

Health Professionals	Number of Participants	*n* = 15
Age	Range	31–71 years
	Mean age (SD)	51.21 (±10.14) years
Gender	Female Male	80% 20%
Role	Cancer Nurse Coordinator	*n* = 3
	Psychosocial support worker or other allied health worker	*n* = 6
	Nurse practitioner	*n* = 1
	Clinical/oncological specialist	*n* = 3
	Clinical psychologist/psychiatrist	*n* = 2
Years of relevant experience	≤10 years	*n* = 5
	≤20 years	*n* = 4
	≤30 years	*n* = 4
	>30 years	*n* = 2
Interview method	Face to face	*n* = 8
	Telephone	*n* = 7
**Parents**	**Number of Participants**	***n* = 11**
Age	Range Mean age (SD)	28–52 years 39.7 (±7.44) years
Gender	Female Male	91% or *n* = 10 9% or *n* = 1
Health status	Patient	5
	Partner	6
Marital status	Married	9
	Separated/Divorced	1
	Widowed	1
Number of children *	1 child	4
	2 children	4
	3 children	2
Age range of children *		1 to 15 years
Cancer type (primary) *	Bowel cancer	2
	Brain	1
	Breast	1
	Burkitts lymphoma	1
	Lymphoma	1
	Melanoma	1
	Non-Hodgkin’s Lymphoma B cell	1
	Lung	1
	Oral	1
Stage * (at time of interview)	II	3
	III	1
	IV	3
	Not reported/remission/deceased	3
Ethnicity	Australian	82% or *n* = 9
	Indonesian	9% or *n* = 1
	Malaysian	9% or *n* = 1
Education	Postgraduate	4
	Tertiary	5
	Other	2
**Children**	**Number of Participants**	***n* = 12**
Age	Range Mean age (SD)	5–17 years 9.46 (±3.43) years
Gender	Female Male	58% or *n* = 7 42% or *n* = 5
Cultural background	Australian	75% or *n* = 9
	Indonesian	17% or *n* = 2
	Malaysian	8% or *n* = 1
Parent with cancer	Mother	50% or *n* = 6 **
	Father	50% or *n* = 6 **
Parent’s primary cancer diagnosis ***	Bowel cancer	2
	Brain	1
	Breast	1
	Burkitt’s lymphoma	1
	Lymphoma	1
	Melanoma	1
	Non-Hodgkin’s Lymphoma B cell	1
	Lung	1
	Oral	1
Stage *** (at time of interview)	II	3
	III	1
	IV	3
	Not reported/remission/deceased ****	3

* One family was represented by two parents; therefore, the responses to these questions were adjusted to account for overrepresentation. ** *n* = 3 sets of siblings (total = 7); therefore, their parent with cancer was counted multiple times. *** Total number of patients included in the study was *n* = 10. **** The deceased patient’s spouse participated in this study.

## Data Availability

Data are available upon request.

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
