# Peer review of "A Conceptual Model Depicting How Children Are Affected by Parental Cancer: A Constructivist Grounded Theory Approach"

_children, 2023, doi:10.3390/children10091507_

Round 1

Reviewer 1 Report

This MS addresses some important mechanisms underlying how children are affected by their parent’s diagnosis of cancer.

It explores breakdowns/barriers in communication/interaction among children, parents, and HPs, in the face of parental cancer.

It also proposes the Alexander’s Children’s Cancer Communication (ACCC) model, which may be used to inform future research studies and intervention development to help in clinical practice.

A minor revision is suggested – some points to consider are presented.

In the Abstract, the abbreviation: ‘HP’ needs to be explained.

Health professionals (HPs) were not detecting children due to barriers that affected their visibility in clinical settings.

In the Introduction section, please change the phrase:’ cancer patients’ to:’ patients with cancer’

The five-year survival rates among Australian cancer patients aged 25 to 49 years are rising [3]

The five-year survival rates among Australian patients with cancer aged 25 to 49 years are rising [3]

In Tab.1.,

‘Must be experienced in providing health care to patients with cancer who have a child or adolescent (up to the age of 18 years) living at home

Could you please explain in more detail what it means: ‘Must be experienced in providing health care to patients with cancer’?

e.g., Medical or Surgical Oncologist, Nurse Practitioner, Nurse Coordinator, Clinical Psychologist/Psychiatrist, Psychosocial support worker, or some other allied health specialists

In Tab.2.,

In the ‘Parents’ section,  could you please indicate the functional status of patients with cancer, e.g., based on the ECOG Scale of Performance Status (PS)?   

Also, may the authors provide a simple graph/table illustrating the unmet needs/challenges of HPs, Parents, & Children, as well as proposed solutions and possible ways to implement them for these 3 groups? [e.g., using some practical/feasible techniques].

Reviewer 2 Report

Dear authors, thank you for your contribution to this important theme.

  1. Introduction:

Line 56: please explain the meaning of the acronyms HPs.

Line 62: I guess it would be clearer for the reader if you could add a phrase to describe the type of intervention performed in the cited studies.

Line 70: I think it would be better to substitute “must” with “should”.

  1. Materials and methods:

Line 105: since it is at the beginning of a phrase. It would be better to substitute the acronyms with the words it refers to. 

Table 2: in the session of table 2 dedicated to parents, you reported the cancer’s stage at the time of interview. How is it possible that the patient at that time was deceased?

Line 140: I guess there is a typing error and you have to remove the comma after HCPs.

Line 148: this phrase needs to be reworded.

Line 172: could you please add some words to explain what NVivo12 is?

4. Discussion:

Line 484-485: here you refer to a communication framework and a communication tool without adding any information that could let the reader who does not know these work to better understand what you mean. I believe it would be better to add some information about these.

Line 508-509: since you have already explained before the acronym “ACC”, here you don’t have to repeat its meaning.

Line 517: at the end of the phrase the parenthesis is missing.  

Reviewer 3 Report

article about A Conceptual Model Depicting How Children Are Affected by 2 Parental Cancer: A Constructivist Grounded Theory Approach

The topics chosen are quite interesting and up to date.

abstracts are prepared in accordance with scientific principles which consist of background, methods, results, conclusions and keywords,

an introductory chapter that contains background, please emphasize what phenomena you encountered so that it becomes your research topic, of course it is supported by relevant data

the research method is good, according to the research title

the results and discussion are quite good, add your opinion to the discussion which is supported by theory. The explanation of the theory is quite detailed

conclusions must answer the research objectives

references are used accordingly

Reviewer 4 Report

Thank you for inviting me to review this manuscript. This manuscript tested a conceptual model regarding how children are affected by parental  concern. Although this paper is generally well-written and contributes to the literature, I do some comments for improvements: 

1. I think you do not need an "aim" section after the introduction but rather you should mention your aim within the introductory section. 

2. Please provide detailed ethical approval information (e.g. the approval number, date of approval, etc.) in the main text. 
